# ANALYTIC DAG CONSTRAINTS FOR DIFFERENTIABLE DAG LEARNING

## ABSTRACT

Recovering underlying Directed Acyclic Graph (DAG) structures from observational data presents a formidable challenge due to the combinatorial nature of the DAG-constrained optimization problem. Recently, researchers have identified gradient vanishing as one of the primary obstacles in differentiable DAG learning and have proposed several DAG constraints to mitigate this issue. By developing the necessary theory to establish a connection between analytic functions and DAG constraints, we demonstrate that analytic functions from the set $\{f(x) = c_0 + \sum_{i=1} c_i x^i | c_0 \geqslant 0; \forall i > 0, c_i > 0; r = \lim_{i \to \infty} c_i/c_{i+1} > 0\}$ can be employed to formulate effective DAG constraints. Furthermore, we establish that this set of functions is closed under several functional operators, including differentiation, summation, and multiplication. Consequently, these operators can be leveraged to create novel DAG constraints based on existing ones. Additionally, we emphasize the significance of the convergence radius $r$ of an analytic function as a critical performance indicator. An infinite convergence radius is susceptible to gradient vanishing but less affected by nonconvexity. Conversely, a finite convergence radius aids in mitigating the gradient vanishing issue but may be more susceptible to nonconvexity. This property can be instrumental in selecting appropriate DAG constraints for various scenarios.

## 1 INTRODUCTION

DAG learning aims to recover Directed Acyclic Graphs (DAGs) from observational data, which is a core problem in many fields, including bioinformatics (Sachs et al., 2005; Zhang et al., 2013), machine learning (Koller & Friedman, 2009), and causal inference (Spirtes et al., 2000). Under certain assumptions (Pearl, 2000; Spirtes et al., 2000), the recovered DAGs can constitute a causal graphical model (CGM) (Koller & Friedman, 2009) and hold causal interpretations.

There are two main categories of DAG learning approaches: constraint-based and score-based methods. Most constraint-based approaches, *e.g.*, PC (Spirtes & Glymour, 1991), FCI (Spirtes et al., 1995; Colombo et al., 2012), rely on conditional independence tests, which typically necessitate a large sample size (Shah & Peters, 2020; Vowels et al., 2021). The score-based approaches, including exact methods based on dynamic programming (Koivisto & Sood, 2004; Singh & Moore, 2005; Silander & Myllymäki, 2006), A* search (Yuan et al., 2011; Yuan & Malone, 2013), and integer programming (Cussens, 2011), as well as greedy methods like GES (Chickering, 2002), model the validity of a graph according to some score function and are often formulated and solved as discrete optimization problems. A key challenge for score-based methods is the super-exponential combinatorial search space of DAGs w.r.t number of nodes(Chickering, 1996; Chickering et al., 2004).

Recently, Zheng et al. (2018) developed a continuous DAG learning approach using Langrange Multiplier methods and a differentiable DAG constraint based on the trace of the matrix exponential of the weighted adjacency matrix. The resulting method, named NOTEARS, demonstrated superior performance in estimating linear DAGs with equal noise variances. Very recently, Zhang et al. (2022) and Bello et al. (2022) suggest that one main issue for NOTEARS and its derivatives, such as Yu et al. (2019), is gradient vanishing for linear DAG model with equal variance. They have thus proposed new continuous DAG constraints by based on geometric series of matrices and log-determinant of matrices.

In fact, most of the proposed DAG constraints can be written in a unified form as in Wei et al. (2020), which shows that for a $d \times d$ adjacency matrices, an order-$d$ polynomial is sufficient and necessary to enforce DAGness. However, from the computational perspective, polynomials of matrices are difficult to compute efficiently. Given the close relation between continuous differentiable functions and analytic functions, we raise the question whether it is possible to use analytic functions to construct DAG constraints, and further more if it is possible to use techniques typically applied to analytic functions to analyze continuous DAG constraints.

The answer is yes. We demonstrate that any analytic function within the class of functions denoted as $\mathcal{F} = \{f | f(x) = c_0 + \sum_{i=0}^{\infty} c_i x^i; c_0 \geqslant 0; c_i > 0, \forall i > 0; \lim_{i \to \infty} c_i / c_{i+1} > 0\}$ can be utilized to formulate Directed Acyclic Graph (DAG) constraints. In fact, the DAG constraints introduced in Zheng et al. (2018), Zhang et al. (2022), and Bello et al. (2022) can all be interpreted as being based on analytic functions from $\mathcal{F}$. Furthermore, we establish that the function class $\mathcal{F}$ remains closed under various function operators, including differentiation, function addition, and function multiplication. Leveraging this insight, we can construct novel DAG constraints based on pre-existing ones. Additionally, we can analyze the performance of these derived DAG constraints using techniques rooted in analytic functions.

## 2 PRELIMINARIES

**DAG Model and Linear SEM**   Given a DAG model $\mathcal{G}$ defined over random vector $\mathbf{x} = [x_1, x_2, \ldots, x_d]^\top$ the corresponding distribution $P(\mathbf{x})$ is assumed to satisfy the Markov assumption (Spirtes et al., 2000; Pearl, 2000). We consider $\mathbf{x}$ to follow a linear Structural Equation Model (SEM):

$$\mathbf{x} = \mathbf{B}^\top \mathbf{x} + \mathbf{e}. \tag{1}$$

Here, $\mathbf{B} \in \mathbb{R}^{d \times d}$ represents the weighted adjacency matrix that characterizes the DAG $\mathcal{G}$, and $\mathbf{e} = [e_1, e_2, \ldots, e_d]^\top$ represents the exogenous noise vector, comprising $d$ independent random variables. To simplify notation, we use $\mathcal{G}(\mathbf{B})$ to denote the graph induced by the weighted adjacency matrix $\mathbf{B}$, and we interchangeably use the terms 'random variables' and 'vertices' or 'nodes'.

We aim to estimate the DAG $\mathcal{G}$ from $n$ i.i.d. examples of $\mathbf{x}$, denoted by $\mathbf{X} \in \mathbb{R}^{n \times d}$. Generally, the DAG $\mathcal{G}$ can be identified only up to its Markov equivalence class under the faithfulness (Spirtes et al., 2000) or the sparsest Markov representation assumption (Raskutti & Uhler, 2018). It has been demonstrated that for linear SEMs with homoscedastic errors, where the noise terms are specified up to a constant (Loh & Bühlmann, 2013), and for linear non-Gaussian SEMs, where no more than one of the noise terms is Gaussian (Shimizu et al., 2006), the true DAG can be fully identified. In our study, we specifically focus on linear SEMs with equal noise variances (Peters & Bühlmann, 2013), where the scale of the data may be either known or unknown. When the scale is known, it is possible to fully recover the DAG. However, in the case of an unknown scale, the DAG may only be identified up to its Markov equivalence class.

**Continuous DAG learning**   In recent years, a series of continuous Directed Acyclic Graph (DAG) learning algorithms Bello et al. (2022); Ng et al. (2020); Zhang et al. (2022); Yu et al. (2021; 2019); Zheng et al. (2018) has been introduced, demonstrating superior performance when applied to linear Structural Equation Models (SEMs) with equal noise variances and known data scale. These methods can be expressed as follows:

$$\operatorname*{argmin}_{\mathbf{B}} S(\mathbf{B}, \mathbf{X}), \text{ s.t. } h(\mathbf{B}) = 0. \tag{2}$$

Here, $S$ is a scoring function, which can take the form of mean square error or negative log-likelihood. The function $h$ is continuous and equal to 0 if and only if $\mathbf{B}$ defines a valid DAG. Previous approaches have employed various techniques, such as matrix exponential (Zheng et al., 2018), log-determinants (Bello et al., 2022), and polynomials (Zhang et al., 2022), to construct the function $h$. However, these methods are known to perform poorly when applied to normalized data since they rely on scale information across variables for complete DAG recovery (Reisach et al., 2021).

## 3 ANALYTIC DAG CONSTRAINTS

In this section, we demonstrate that the diverse set of continuous DAG constraints proposed in previous work can be unified through the use of analytic functions. We will begin by offering a brief

introduction to analytic functions and then illustrate how they can be employed to establish DAG constraints.

## 3.1 Analytic functions as DAG constraints

In mathematics, a power series

$$f(x) = c_0 + \sum_{i=1}^{\infty} c_i x^i, \tag{3}$$

which converges for $|x| \leqslant r = \lim_{i \to \infty} |c_i/c_{i+1}|$, defines an analytic function $f$ on the interval $(-r, r)$, and $r$ is known as the convergence radius. When we replace $x$ with a matrix $\mathbf{A}$, we obtain an analytic function $f$ of a matrix as follows:

$$f(\mathbf{A}) = c_0 \mathbf{I} + \sum_{i=1}^{\infty} c_i \, \mathbf{A}^i, \tag{4}$$

where $\mathbf{I}$ is the identity matrix. Equation (4) will converge if the largest absolute value of eigenvalues of $\mathbf{A}$, known as the spectral radius and denoted by $\rho(\mathbf{A})$, is smaller than $r$.

We are particularly interested in the following specific class of analytic functions

$$\mathcal{F} = \{f | f(x) = c_0 + \sum_{i=1}^{\infty} c_i x^i; c_0 \geqslant 0; \forall i > 0, c_i > 0; \lim_{i \to \infty} c_i/c_{i+1} > 0\}, \tag{5}$$

as any analytic function belongs to $\mathcal{F}$ can be applied to construct a continuous DAG constraint.

**Proposition 1.** *Let $\tilde{\mathbf{B}} \in \mathbb{R}_{\geq 0}^{d \times d}$ with $\rho(\tilde{\mathbf{B}}) \leqslant r$ be the weighted adjacency matrix of a directed graph $\mathcal{G}$, and let $f$ be an analytic function in the form of* (3)*, where we further assume $\forall i > 0$ we have $c_i > 0$, then $\mathcal{G}$ is acyclic if and only if*

$$\mathrm{tr}\left[f(\tilde{\mathbf{B}})\right] = c_0 d. \tag{6}$$

An interesting property the DAG constraint (6) is that its gradients can also be represented as transpose of an analytic function as follows.

**Proposition 2.** *There exists some real number $r$, where for all $\{\tilde{\mathbf{B}} \in \mathbb{R}_{\geq 0}^{d \times d} | \rho(\tilde{\mathbf{B}}) < r\}$, the derivative of $\mathrm{tr}\left[f(\tilde{\mathbf{B}})\right]$ w.r.t. $\tilde{\mathbf{B}}$ is*

$$\nabla_{\tilde{\mathbf{B}}} \mathrm{tr}\left[f(\tilde{\mathbf{B}})\right] = \left[\nabla_x f(x)|_{x=\tilde{\mathbf{B}}}\right]^{\top}. \tag{7}$$

It is notable that for a $d \times d$ adjacency matrix $\tilde{\mathbf{B}}$, an order-$d$ polynomial of $\tilde{\mathbf{B}}$ is sufficient to enforce DAGness (Wei et al., 2020; Ng et al., 2022). Computing matrix polynomials efficiently is highly nontrivial (Higham, 2008). For matrix analytic functions such as exponentials or logarithms, however, efficient algorithms exist (Higham, 2008).

The connection between matrix analytic functions and real analytic functions means that various properties of the matrix function can be obtained from a simple real valued function. To pursue DAG constraints with better computational efficiency, we seek an analytic function whose derivative can be represented by itself to reduce the computation of different analytic functions. If a function do have such property, various intermediate results can be saved for future computation of gradients. The exponential function $\exp(x)$ with $\partial \exp(x)/\partial x = \exp(x)$, is a natural contender, and this leads to the well-known exponential function based DAG constraints (Zheng et al., 2018)

$$\textbf{Constraints: } \mathrm{tr}\left[\exp(\tilde{\mathbf{B}})\right] = \sum_{i=0}^{\infty} \tilde{\mathbf{B}}^i/i! = d, \quad \textbf{Gradient: } \nabla_{\tilde{\mathbf{B}}} \exp(\tilde{\mathbf{B}}) = \exp(\tilde{\mathbf{B}})^{\top}, \tag{8}$$

which will converge for any $\tilde{\mathbf{B}}$.

Recently Bello et al. (2022) and Zhang et al. (2022) have suggested that exponential based DAG constraints suffers from gradient vanishing. One cause of gradient vanishing arises from the

small coefficients of high order terms. The convergence radius for the exponential is $\infty$, that is $\lim_{i\to\infty} |c_i/c_{i+1}| = \lim_{i\to\infty} |(i+1)!/i!| = \infty$, which suggests that, compared to the lower order terms, the higher order terms contribute almost nothing in the DAG constraints.

Due to the fact that the adjacency matrix of a DAG must form a nilpotent matrix, we do not need a function with infinite convergence radius. Instead, we can use an analytic function with finite convergence radius $r = \lim_{i\to\infty} |c_i/c_{i+1}| < \infty$. Thus by using a sequence $c_i$ with geometric progression $c_i = 1/s^{i-1}$ or harmonic-geometric progression $c_i = 1/is^{i-1}$ we can obtain two analytic functions,

$$f_{inv}^s(x) = (s-x)^{-1} = \sum_{i=0}^{\infty} x^i/s^{i-1}, \quad f_{log}^s(x) = -s\log(s-x) = \sum_{i=1}^{\infty} \frac{x^i}{is^{i-1}}. \tag{9}$$

Then by our Proposition 1 and Proposition 2, two dag constraints can be obtained as follows:

$$\textbf{Constraints: } \operatorname{tr} f_{inv}^s(\tilde{\mathbf{B}}) = d, \qquad \textbf{Gradient: } \nabla_{\tilde{\mathbf{B}}} \operatorname{tr} f_{inv}^s(\tilde{\mathbf{B}}) = [f_{inv}^s(\tilde{\mathbf{B}})^2]^\top, \tag{10a}$$

$$\textbf{Constraints: } \operatorname{tr} f_{log}^s(\tilde{\mathbf{B}}) = 0, \qquad \textbf{Gradient: } \nabla_{\tilde{\mathbf{B}}} \operatorname{tr} f_{log}^s(\tilde{\mathbf{B}}) = [f_{inv}^s(\tilde{\mathbf{B}})]^\top, \tag{10b}$$

where a truncated version of $f_{inv}^s$ is applied in Zhang et al. (2022), and the $f_{log}^s$ based constraints are equivalent to those in Bello et al. (2022). One key difference between Zhang et al. (2022); Bello et al. (2022) and the exponential-based DAG constraints (Zheng et al., 2018) is their finite convergence radius, which requires an additional constraints $\rho(\tilde{\mathbf{B}}) < s$. Meanwhile, the adjacency matrix of a DAG must be nilpotent, and thus its spectral radius must be 0. In this case, such additional constraints would not affect the feasible set.

## 3.2 Constructing DAG constraints by functional operator

One can easily observe a coincidence between $f_{log}$ and $f_{inv}$ as follows,

$$\frac{\partial f_{log}^s(x)}{\partial x} = f_{inv}^s(x), \quad f_{log}^s(x) = \int f_{inv}^s(t)\mathbf{d}t + C, \tag{11}$$

which suggests that it may be possible to derive a group of DAG constraints from an analytic function by applying integration or differentiation. This is because derivatives of any order of an analytic function is also analytic. More formally, if a function is analytic at some point $x_0$, then its $n^{\text{th}}$ derivative for any integer $n$ exists and is also analytic at $x_0$. Thus we can derive DAG constraints from any $f \in \mathcal{F}$ as follows.

**Proposition 3.** *Let $f(x) = c_0 + \sum_{i=1}^{\infty} c_i x^i \in \mathcal{F}$ be analytic on $(-r, r)$, and let $n$ be arbitary integer larger than 1, then $\tilde{\mathbf{B}} \in \mathbb{R}_{\geqslant 0}^{d\times d}$ with spectral radius $\rho(\hat{\mathbf{B}}) \leqslant r$ forms a DAG if and only if*

$$\operatorname{tr}\left[\left.\frac{\partial^n f(x)}{\partial x^n}\right|_{x=\tilde{\mathbf{B}}}\right] = n!c_n. \tag{12}$$

The above proposition suggests that the differential operator can be applied to an analytic function to form a new DAG constraints. Besides the differential operator, the addition and multiplication of analytic functions can also be applied to generate new DAG constraints. That is

**Proposition 4.** *Let $f_1(x) = c_0^1 + \sum_{i=1}^{\infty} c_i^1 x^i \in \mathcal{F}$, and $f_2(x) = c_0^2 + \sum_{i=1}^{\infty} c_i^2 x^i \in \mathcal{F}$. Then for an adjancency matrix $\tilde{\mathbf{B}} \in \mathbb{R}_{\geqslant 0}^{d\times d}$ with spectral radius $\rho(\tilde{\mathbf{B}}) \leqslant \min(\lim_{i\to\infty} c_i^1/c_{i+1}^1, \lim_{i\to\infty} c_i^2/c_{i+1}^2)\}$, the following three statements are equivalent:*

1. *$\tilde{\mathbf{B}}$ forms a DAG;*

2. *$\operatorname{tr}[f_1(\tilde{\mathbf{B}}) + f_2(\tilde{\mathbf{B}})] = (c_0^1 + c_0^2)d$;*

3. *$\operatorname{tr}[f_1(\tilde{\mathbf{B}})f_2(\tilde{\mathbf{B}})] = c_0^1 c_0^2 d$.*

Particularly for $f_{log}^s(x)$ and $f_{inv}^s(x)$, due to the specific property of $f_{inv}^s(x)$, we have

$$\frac{\partial^{n+1} f_{log}^s(x)}{\partial x^{n+1}} = \frac{\partial^n f_{inv}^s(x)}{\partial x^n} \propto (s-x)^{-(n+1)} = [f_{inv}^s(x)]^{n+1}. \tag{13}$$

Here we can see another very good property of the function $1/(s-x)$, that is its $n^{\text{th}}$ derivative is proportional to the order-$(n+1)$ power of $1/(s-x)$. With this property, $(\mathbf{I} - \tilde{(\mathbf{B})}/s)^{-1}$ can be cached during the evaluation of DAG constraints, and later they can be applied for efficient evaluation of gradients. Furthermore, the gradients of the DAG constraints will also increase as $n$ increases.

**Proposition 5.** *Let $n$ be any positive integer, the adjacency matrix $\tilde{\mathbf{B}} \in \{\hat{\mathbf{B}} \in \mathbb{R}_{\geqslant 0}^{d \times d} | \rho(\hat{\mathbf{B}}) \leqslant s\}$ forms a DAG if and only if*

$$\text{tr}\left[(\mathbf{I} - \tilde{\mathbf{B}})^{-n}\right] = d.$$

*Furthermore, the gradients of the DAG constraints satisfies that $\forall \tilde{\mathbf{B}} \in \{\hat{\mathbf{B}} \in \mathbb{R}_{\geqslant 0}^{d \times d} | \rho(\hat{\mathbf{B}}) \leqslant s\}$*

$$\|\nabla_{\tilde{\mathbf{B}}} \text{tr}(\mathbf{I} - \tilde{\mathbf{B}})^{-n}\| \leqslant \|\nabla_{\tilde{\mathbf{B}}} \text{tr}(\mathbf{I} - \tilde{\mathbf{B}})^{-n-k}\|,$$

*where $k$ is an arbitrary positive integer, and $\|\cdot\|$ an arbitrary matrix norm.*

As gradient vanishing is one main challenges for differentiable DAG learning, we may prefer larger $n$ to get better performance in practice. The performance gap between different $n$ may not be significant, however, because the convergence radius $s$ is constant, which means that in the limiting case the contribution from higher order terms and lower orders are in the same ratio. Moreover, the performance gap between this series of DAG constraints and the exponential-based approaches may be large due to the infinite convergence radius of the exponential function.

### 3.3 OVERALL OPTIMIZATION FRAMEWORK

The DAG constraints above are applicable only to positive adjacency matrices, so we use the Hadamard product to map a real adjacency matrix to a positive one. Thus Equation (2) becomes:

$$\underset{\mathbf{B}}{\arg\min}\, S(\mathbf{B}, \mathbf{X}), \text{ s.t. } \text{tr}f(\mathbf{B} \odot \mathbf{B}) = c_0 d, \rho(\mathbf{B} \odot \mathbf{B}) < r, \tag{14}$$

where the analytic function $f(x) = c_0 + \sum_{i=1}^{\infty} x^i \in \mathcal{F}$, and $\odot$ denotes the Hadamard product.

In our work, we choose to use the path-following approach with an $\ell_1$ regularizer, as in Bello et al. (2022). This is because in the Lagrange approaches applied in Zhang et al. (2022); Yu et al. (2021); Zheng et al. (2018); Yu et al. (2019), the Lagrangian multiplier must be set to very large value to enforce DAGness, which may result in numerical instability. In the path-following approach, instead of using large Lagrangian multipliers, a small coefficients are added to the score function $S$ as follows[1]

$$\underset{\mathbf{B}}{\arg\min}\, \mu[S(\mathbf{B}, \mathbf{X}) + \lambda_1 \|\mathbf{B}\|_1] + \text{tr}f(\mathbf{B} \odot \mathbf{B}), \text{ s.t. } \rho(\mathbf{B} \odot \mathbf{B}) < r, \tag{15}$$

where $\lambda_1$ is the user-specified weight for the $\ell_1$ regularizer. For the additional constraints $\rho(\mathbf{B} \odot \mathbf{B}) < r$, with properly chosen initial value and step-length, it can usually be satisfied. Also it is notable that $\|\mathbf{B}\|_1 < r$ is a sufficient condition for $\rho(\mathbf{B} \odot \mathbf{B}) < r$, and thus the sparsity constraints also encourage this condition to be satisfied. Based on Bello et al. (2022), we implemented a path-following shown in Algorithm 1.

---

**Algorithm 1** Path following algorithm

---

**Input:** The observational data $\mathbf{X} \in \mathbb{R}^{n \times d}$; a differentiable score function $S$; an analytic function $f \in \mathcal{F}$; the weight $\lambda_1$ for $\ell_1$ regularizer; an initial coefficients $\mu_0$; a decay factor $\alpha$; Number of iterations $T$;
**Output:** Estimated $\mathbf{B}$
1: $i \leftarrow 0, \mu \leftarrow \mu_0, \mathbf{B}_0 = \mathbf{0}$
2: **while** $i < T$ **do**
3:     Obtain $\mathbf{B}_{i+1}$ by solving $\arg\min_{\mathbf{B}} \mu[S(\mathbf{B}, \mathbf{X}) + \lambda_1 \|\mathbf{B}\|_1] + \text{tr}f(\mathbf{B} \odot \mathbf{B})$ by gradient based approaches using $\mathbf{B}_i$ as starting point.
4:     $\mu \leftarrow \mu \times \alpha$
5:     $i \leftarrow i + 1$
6: **end while**
7: **Output** $\mathbf{B}_T$

---

[1] the constant $c_0 d$ can be dropped because $\text{tr}f(\mathbf{B} \odot \mathbf{B})$ is bounded below by $c_0 d$, detailed derivation is provided in the supplementary file.

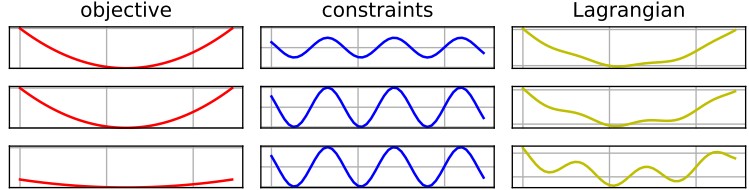

Figure 1: Illustration of different objective, DAG constraints and Lagrangian. The differentiable DAG learning problem can be viewed as optimize over a Lagrangian which is a summation objective plus a invex constraints. For DAG learning with known data scale, the objective function provides sufficient information (illustrated in **top two rows**), and DAG constraints with larger gradients to is preferred to provide a sharper curve (**middle rows**). Meanwhile, for DAG learning problem with unknown scale (**bottom row**), the information from objective is quite limited, DAG constraints with smaller gradients are preferred to avoid highly non-convex Lagrangian.

**Choosing a score function.** For linear SEMs with equal variances, choosing mean square error (MSE) as a score function should be sufficient to recover the true DAG. In this case, the optimization problem becomes:

$$\underset{\mathbf{B}}{\arg\min} \mu[\|\mathbf{X} - \mathbf{X}\mathbf{B}\|_F^2 + \lambda_1\|\mathbf{B}\|] + \mathrm{tr} f(\mathbf{B} \odot \mathbf{B}), \tag{16}$$

then for any $f \in \mathcal{F}$ we can apply the same path-following procedure as Bello et al. (2022) to estimate the DAG from observational data $\mathbf{X}$.

However if there is unknown scaling applied to the data $\mathbf{X}$, using MSE will not guarantee recovery of the true DAG, and its performance is usually much worse than in the known scale case. We can use maximum likelihood based losses such as the GOLEM-NV loss, which is specifically designed for data with unknown scale (Ng et al., 2020). However, the performance of these approaches is still worse than that of constraint-based approaches such as PC (Spirtes & Glymour, 1991; Ng et al., 2023). This is partially because the invex property of the DAG constraints (Bello et al., 2022). The DAG constraints have many local minimal, whose projective is similar to the plot in Figure 1. Thus if the objective value can not provide enough information, the final Lagrangian in Equation (15) becomes highly nonconvex. We observed that by encouraging the correlation coefficients of the residuals between each $e_i$ and $e_j$ to be small(recall $\mathbf{e} = \mathbf{x} - \mathbf{B}^\top \mathbf{x}$), the performance of continuous DAG learning can be improved to a similar level as that of constraint-based approaches.

**Choosing DAG constraints for different scenarios** The optimization problem (16) can be viewed as a convex objective plus one invex constraints. The non-convexity of the DAG constraint term can be roughly measured by the Frobenius norm, or the largest off-diagonal entry of its Hessian. We provide the Hessian in Equation (33) and show that the Frobenius norm/the largest off-diagonal is closely related the coefficients $c_i$. Larger $c_i$ will cause larger Frobenius norm/the largest off-diagonal of the Hessian, and thus may cause more severe non-convexity. In this case, for data with known scale, the objective term contains enough information and thus the non-convexity may not be a serious issue. Thus DAG constraints with finite convergence radius is preferred to provide larger gradients. Meanwhile, Meanwhile, for DAG learning problem with unknown scale, the information from objective is quite limited. Thus we may prefer DAG constraints with infinite convergence radius to escape from the non-convexity. More discussion is provided in Appendix B.

## 4 EXPERIMENTS

In the experiment, we compared the performance of different analytic DAG constraints in the same path-following optimization framework. We implemented the path-following algorithm (provided in Algorithm 1) using PyTorch (Paszke et al., 2019) based on the path-following optimizer in Bello et al. (2022). For analytic DAG constraints with infinite convergence radius, we consider the exponential based DAG constraints. For analytic DAG constraints with finite convergence radius, we consider the follow 4 different DAG constraints generated by the differentiation operator or multiply operator:

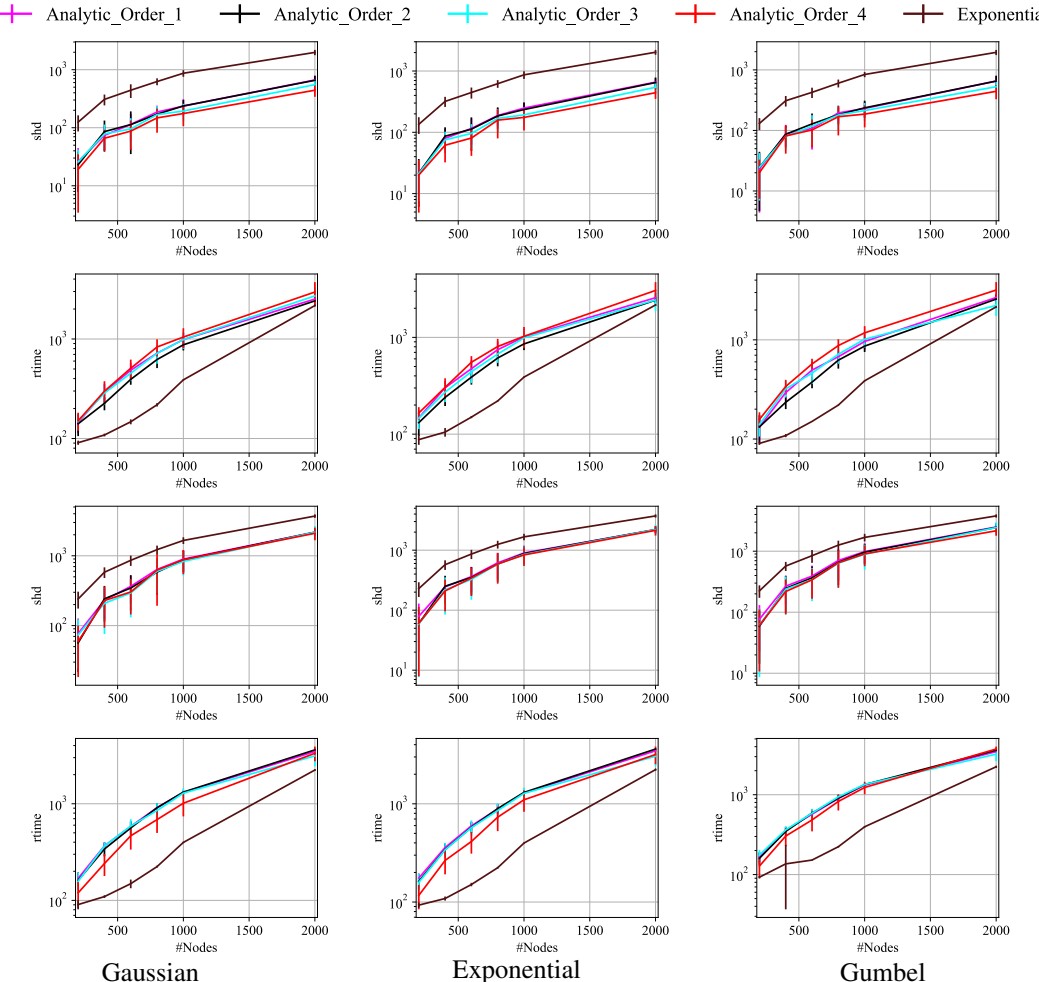

Figure 2: DAG learning performance of different DAG constraints on ER3 (**Top 2 rows**) and ER4 (**Bottom 2 rows**) graphs. In terms of SHD (shown in the **First** and the **Third** row, the lower the better), all DAG constraints with finite convergence radius performs better than the Exponential based approach, and the Order-4 (*i.e.* the one with largest Gradient norm) usually performs better than others. In terms of running time (shown in the Second and the **Forth** row, the lower the better), the Exponential based approach has the shortest running time, and all other approach has similar running time.

- Order-1: $\mathrm{tr} f_{\log}^s (\mathbf{B} \odot \mathbf{B}) = 0$;
- Order-2: $\mathrm{tr} f_{inv}^s (\mathbf{B} \odot \mathbf{B} / s) = d$;
- Order-3: $\mathrm{tr} [f_{inv}^s (\mathbf{B} \odot \mathbf{B} / s)]^2 = d$;
- Order-4: $\mathrm{tr} [f_{inv}^s (\mathbf{B} \odot \mathbf{B} / s)]^3 = d$.

As stated in previous section, the gradients of these four DAG constraints will be larger than the gradients of the exponential based DAG constraints. We compare the performance of these DAG constraints using two different settings: linear SEM with known ground truth scale and with unknown ground truth scale. We also compare these methods with constraint based PC (Spirtes & Glymour, 1991) algorithm and score based combinatorial search algorithm GES (Chickering, 2002) implemented by Kalainathan et al. (2020).

## 4.1 LINEAR SEM WITH KNOWN GROUND TRUTH SCALE

For linear SEM with known ground truth scale, our experimental setting is similar to Bello et al. (2022); Zhang et al. (2022); Zheng et al. (2018). We generated two different type of random graphs: ER (Erdős-Rényi) and SF (Scale-Free) graphs with different number of expected edges. We use ER$n$ (SF$n$) to denote graphs with $d$ nodes and $nd$ expected edges. Edge weights generated from uniform distribution over the union of two intervals $[-2, -0.5] \cup [0.5, 2.0]$ are assigned to each edge to form a weighted adjacency matrix $\mathbf{B}$. Then $n$ samples are generated from the linear SEM $\mathbf{x} = \mathbf{B} \mathbf{x} + \mathbf{e}$ to form an $n \times d$ data matrix $\mathbf{X}$, where the noise $\mathbf{e}$ are i.i.d. sampled from Gaussian, Exponential,

Table 1: DAG learning performance of different algorithms on linear SEM with unknown ground truth scale (measuredby SHD of CPDAGs denoted by SHDC, the lower the better). We compare differential DAG learning approaches with conditional independent test based PC (Spirtes & Glymour, 1991) algorithm and score based GES (Chickering, 2002) algorithm. The result is reported in the format of average$\pm$ standard derivation gathered from 10 different simulations.

|  | PC | GE | Exponential | Order-1 | Order-2 |
|---|---|---|---|---|---|
| 15-node ER2 (SHDC) | 24.9$\pm$8.8 | 30.2$\pm$17.4 | 23.2 $\pm$ 10.1 | 23.8 $\pm$ 10.6 | 25.6 $\pm$ 9.4 |
| 50-node ER1 (SHDC) | 7.8$\pm$3.4 | 22.9$\pm$6.9 | 11.6$\pm$4.3 | 14.2$\pm$8.5 | 21.8$\pm$9.3 |

or Gumbel distribution. We set the sample size $n = 1000$ and consider 6 different number of nodes $d = 200, 400, 600, 800, 1000, 2000$. For each setting, we conduct 20 random simulations to obtain an average performance. All these experiments are done using A100 GPU, and all computations are done in double precision.

The result on ER3 and ER4 graphs is shown in Figure 2. In both ER3 and ER4 the constraints with finite convergence radius performs better than the one with infinite convergence radius in terms of Structural Hamming Distance (SHD). For ER3 graph, the Order-4 constraints usually performs slightly better than the others. For ER4 graphs, all constraints with finite convergence radius have similar performance in terms of SHD, and the Order-4 constraints performs slightly better than others with node number 2000. For the running time, all algorithms have similar running time and the one with exponential based constraints run slightly faster than others. Previously Bello et al. (2022) reported that the expoential constraints based approach NOTEARS (Zheng et al., 2018) runs much more slower than other scalable methods, which is because of the slow optimization methods in NOTEARS.

The result on SF3 and SF4 graphs is shown in Figure 3. In terms of SHD, the constraints with finite convergence radius performs slightly better than the one with infinite convergence radius, and all constraints with finite convergence radius achieves very similar performance. In terms of running time, ther performance of all approaches are very similar. This results suggest that on ER graph, the gradient vanishing problem may be more serius than the SF graph, and thus for ER graphs higher order constraints with larger gradient may be preferred. Meanwhile for SF graphs any analytic constraints with finite convergence radius would results similar performance.

We also conducted experiments on smaller ER and SF graphs and the results is provided in the supplementary file.

## 4.2 LINEAR SEM WITH UNKNOWN GORUND TRUTH SCALE

For linear SEM with unknown ground truth scale, we applied the same data generation process as the linear SEM with known ground truth scale and Gaussian noise, but normalize the generate data $\mathbf{X}$ to zero mean with unit variance. In this normalization procedure, the scale information of variable is stripped from data. Particularly for gaussian noise, in this case the true DAG is not identifiable we may only identify it to a Markov Equivalent Class. We considered two cases: 15-nodes ER2 graphs and 50-node ER1 graphs. For the 15-nodes ER2 graphs, 200 observational samples are generated and for the 50-node ER1 graphs, 1000 observational samples are generated. For each case, we conduct 10 different simulation to report the average performance. Here instead of using MSE loss as score, we further add the $F$-norm of the correlation coefficients matrix of residuals $\mathbf{X} - \mathbf{X}\mathbf{B}$ as stated in previous section.

In this experiment, for continuous DAG learnign approaches we first learn a DAG from data and then convert the DAG to completed partially directed acyclic graph(CPDAG) to measure the SHD between the estimated CPDAG and the ground truth CPDAG. The results is shown in Table 2. While previously it is observed by Ng et al. (2023); Reisach et al. (2021) that the performance of continuous DAG learning is often far worse than PC(Spirtes & Glymour, 1991) and GES(Chickering, 2002). We show that it may be possible for continuous learning approaches to achieve similar performance as PC and GES. The result is shown in Table 2, where on 15-node ER2 graphs, the exponential based DAG constraints achieves similar performance as PC, while for 50-node ER1 graph, the result of exponential DAG constraints is slightly worse than PC. Furthermore, here the performance of DAG constraints with finite converge radius performances worse than the exponential based approach.

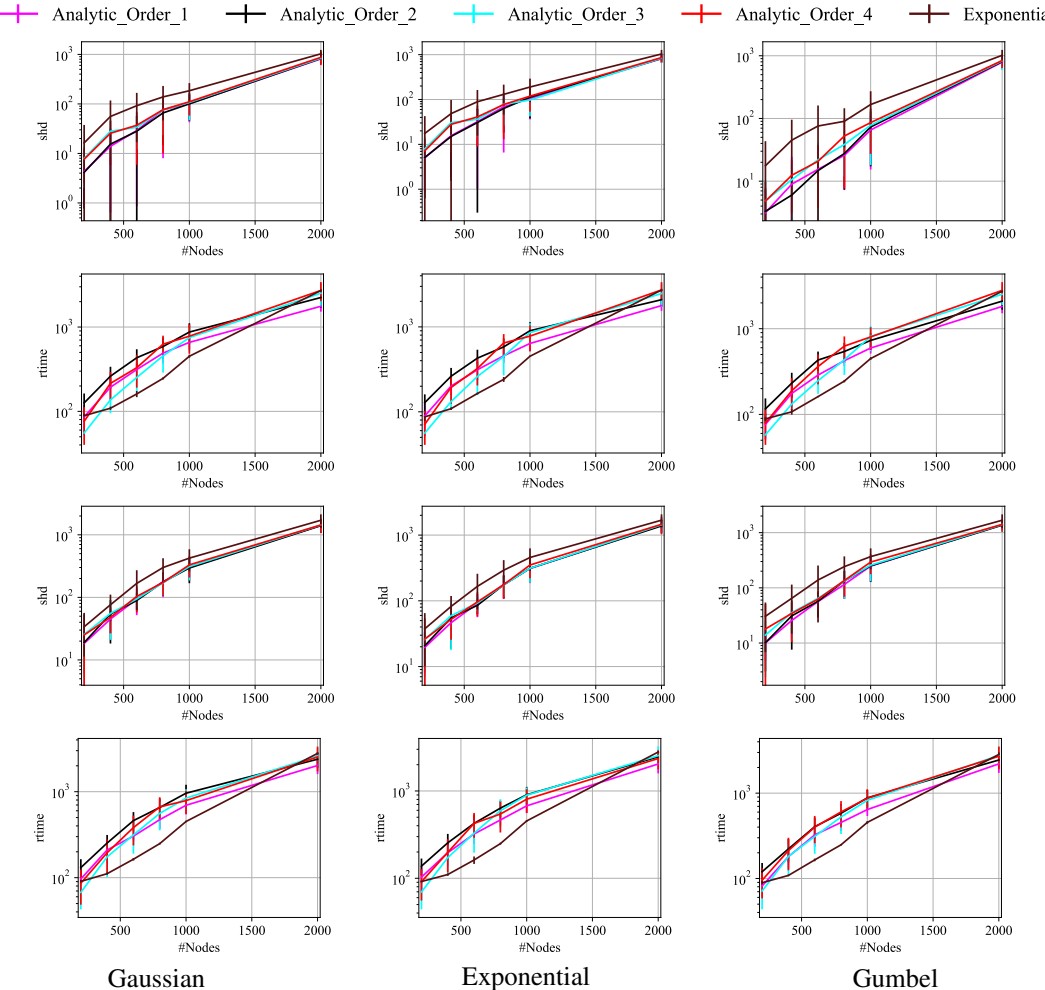

Figure 3: DAG learning performance of different DAG constraints on SF3 (**Top 2 rows**) and SF4 (**Bottom 2 rows**) graphs.In terms of SHD (shown in the **First** and the **Third** row, the lower the better), all DAG constraints with finite convergence radius has similar performance and they performs slightly better than the Exponential based approach. In terms of running time (shown in the Second and the **Forth** row, the lower the better), all approaches have similar running time.

One possible reason is that for these DAG learning problem, instead of gradient vanishing, the non-convexity of the optimization problem may be the main issue. The higher order terms, although diminishes in the exponential constraints, however this also makes the constraints less non-convex.

# 5 CONCLUSION

The continuous differentiable DAG constraints play an important role in the continuous DAG learning algorithms. We show that many of these DAG constraints can be formulated using analytic functions. Those analytic function based DAG constraints can be categorized into two classes: analytic functions with infinite convergence radius and ones with finite convergence radius. Viewing analytic function as power series $c_0 + \sum_{i=1}^{\infty} c_i x_i$, a main difference between the former ones and the later ones is if $\lim_{i\to\infty} c_i/c_{i+1}$ diverges to infinity or not. This results in different behavior of DAG constraints. For the former one, $\lim_{i\to\infty} c_i/c_{i+1}$ diverges to infinity, which indicates it must suffers more from gradient vanishing due to the small coefficients on higher order terms. However, it may be less nonconvex as the coefficients of highly nonconvex higher order terms are smaller. On the contrary, the constraints based on analytic functions with finite convergence radius suffers less from gradient vanishing, but more from nonconvexity.

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

# Appendices

## A  PROOF OF PROPOSITIONS

Our proof are also based on the well-known properties of analytic functions listed as follows:

1. Let $f_1(x)$, $f_2(x)$ be analytic functions on $(-r_1, r_1)$ and $(-r_2, r_2)$, then $f_1(x) + f_2(x)$ and $f_1(x)f_2(x)$ are analytic functions on $(-\min(r_1, r_2), \min(r_1, r_2))$;

2. Let $f(x)$ be analytic function on $(-r, r)$, then $\partial f(x)/\partial x$ is an analytic function on $(-r, r)$.

### A.1  LEMMAS REQUIRED FOR PROOFS

**Lemma 6.** *Let $\tilde{\mathbf{B}} \in \mathbb{R}_{\geqslant 0}^{d \times d}$ be the weighted adjacency matrix of a graph $\mathcal{G}$ with $d$ vertices, $\mathcal{G}$ is a DAG if and only if $\tilde{\mathbf{B}}^d = \mathbf{0}$.*

*Proof.* See Proposition 3.1 of Zhang et al. (2022). $\qquad\square$

**Lemma 7.** *Let $\tilde{\mathbf{B}} \in \mathbb{R}_{\geqslant 0}^{d \times d}$ be the weighted adjacency matrix of a graph $\mathcal{G}$ with $d$ vertices, $\mathcal{G}$ is a DAG if and only*

$$\mathrm{tr}(\sum_{i=1}^{d} c_i \tilde{\mathbf{B}}^i) = 0,$$

*where $c_i > 0 \forall i$.*

*Proof.* See Wei et al. (2020). $\qquad\square$

### A.2  PROOF OF PROPOSITION 1

**Proposition 1.** *Let $\tilde{\mathbf{B}} \in \mathbb{R}_{\geq 0}^{d \times d}$ with $\rho(\tilde{\mathbf{B}}) \leqslant r$ be the weighted adjacency matrix of a directed graph $\mathcal{G}$, and let $f$ be an analytic function in the form of (3), where we further assume $\forall i > 0$ we have $c_i > 0$, then $\mathcal{G}$ is acyclic if and only if*

$$\mathrm{tr}\left[f(\tilde{\mathbf{B}})\right] = c_0 d.$$

*Proof.* Without loss of generality, assume that $f$ can be formulated as:

$$f(x) = c_0 + \sum_{i=1}^{\infty} c_i x^i; \forall i, c_i > 0; \lim_{i \to \infty} c_i/c_{i+1} > 0. \tag{17}$$

First if $\mathcal{G}$ is acyclic, by Lemma 6 we must have

$$\tilde{\mathbf{B}}^k = \mathbf{0} \forall k \geqslant d, \tag{18}$$

which also indicates that $\rho(\tilde{\mathbf{B}}) = 0$. Thus we have

$$
\begin{aligned}
\mathrm{tr}\left[f(\tilde{\mathbf{B}})\right] =& \mathrm{tr}\left[c_0\mathbf{I} + \sum_{i=1}^{d} c_i \tilde{\mathbf{B}}^i + \underbrace{\sum_{i=d+1}^{\infty} c_i \tilde{\mathbf{B}}^i}_{\text{\color{red}Equals } \mathbf{0}\text{, By Lemma 6}}\right]\\
=& \mathrm{tr}[c_0\mathbf{I}] + \underbrace{\mathrm{tr}\left[\sum_{i=1}^{d} c_i \tilde{\mathbf{B}}^i\right]}_{\text{\color{red}Equals 0, By Lemma 7}}\\
=& c_0 d.
\end{aligned}
\tag{19}
$$

On the other hand, if $\mathrm{tr}\left[f(\tilde{\mathbf{B}})\right] = c_0 d$, we must have that

$$\mathrm{tr}\left[\sum_{i=1}^{\infty} c_i \tilde{\mathbf{B}}^i\right] = 0.$$

By the fact all entries of $\tilde{\mathbf{B}}$ are positive, we have that

$$0 \leqslant \mathrm{tr}\left[\sum_{i=1}^{d} c_i \tilde{\mathbf{B}}^i\right] \leqslant \left[\sum_{i=1}^{\infty} c_i \tilde{\mathbf{B}}^i\right] = 0. \tag{20}$$

Then we must have

$$\mathrm{tr}\left[\sum_{i=1}^{d} c_i \tilde{\mathbf{B}}^i\right] = 0.$$

Finally by Lemma 7 we have that $\mathcal{G}$ is a DAG. $\qquad\square$

## A.3 Proof of Proposition 2

In all the paper, we consider analytic functions $f$ from the functional class $\mathcal{F}$ defined in (5).

**Proposition 2.** *There exists some real number $r$, where for all $\{\tilde{\mathbf{B}} \in \mathbb{R}_{\geqslant 0}^{d \times d} | \rho(\tilde{\mathbf{B}}) < r\}$, the derivative of $\mathrm{tr}\left[f(\tilde{\mathbf{B}})\right]$ w.r.t. $\tilde{\mathbf{B}}$ is*

$$\nabla_{\tilde{\mathbf{B}}} \mathrm{tr}\left[f(\tilde{\mathbf{B}})\right] = \left[\nabla_x f(x)|_{x=\tilde{\mathbf{B}}}\right]^{\top}.$$

*Proof.* Without loss of generality, assume that $f$ can be formulated as:

$$f(x) = c_0 + \sum_{i=1}^{\infty} c_i x^i; \forall i, c_i > 0; \lim_{i \to \infty} c_i / c_{i+1} > 0. \tag{21}$$

For some $i$ by basic matrix differentiation we have

$$\frac{\partial \mathrm{tr}\tilde{\mathbf{B}}^i}{\partial \tilde{\mathbf{B}}} = (i\,\mathbf{B}^{i-1})^{\top}, \tag{22}$$

and then by the properties of power series we have

$$\begin{aligned}
\nabla_{\tilde{\mathbf{B}}} \mathrm{tr}\left[f(\tilde{\mathbf{B}})\right] &= \nabla_{\tilde{\mathbf{B}}} \mathrm{tr}\left[c_0 \mathbf{I} + \sum_{i=1}^{\infty} c_i \tilde{\mathbf{B}}^i\right] \\
&= \sum_{i=1}^{\infty} \nabla_{\tilde{\mathbf{B}}} \mathrm{tr} c_i \tilde{\mathbf{B}}^i \\
&= \left[\sum_{i=1}^{\infty} c_i i \tilde{\mathbf{B}}^{i-1}\right]^{\top} \\
&= \left[\left.\sum_{i=1}^{\infty} c_i i x^{i-1}\right|_{x=\tilde{\mathbf{B}}}\right]^{\top} = \left[\nabla_x f(x)|_{x=\tilde{\mathbf{B}}}\right]^{\top},
\end{aligned} \tag{23}$$

where we can exchange $\nabla$ and $\sum_{i=1}^{\infty}$ because after the exchanging the new power series will still converge (by properties of analytic functions). $\qquad\square$

## A.4 Proof of Proposition 3

**Proposition 3.** *Let $f(x) = c_0 + \sum_{i=1}^{\infty} c_i x^i \in \mathcal{F}$ be a analytic function on $(-r, r)$, and let $n$ be arbitary integer larger than 1, then $\tilde{\mathbf{B}} \in \mathbb{R}_{\geqslant 0}^{d \times d}|$ with spectral radius $\rho(\hat{\mathbf{B}}) \leqslant r$ forms a DAG if and only if*

$$\mathrm{tr}\left[\left.\frac{\partial^n f(x)}{\partial x^n}\right|_{x=\tilde{\mathbf{B}}}\right] = n! c_n.$$

*Proof.* By properties of analytic functions, the $n^{\text{th}}$ order derivative of an analytic function $f(x)$ on $(-r, r)$ is still an analytic function on $(-r, r)$. Particularly for $f(x) = c_0 + \sum_{i=1}^{\infty} c_i x^i \in \mathcal{F}$, we have

$$
\begin{aligned}
\frac{\partial^n f(x)}{\partial x^n} &= \sum_{i=1}^{\infty} \frac{\partial^n c_i x^i}{\partial x^n} \\
&= \sum_{i=n}^{\infty} \frac{\partial^n c_i x^i}{\partial x^n} \\
&= \sum_{i=n}^{\infty} \left[ c_i x^{i-n} \prod_{k=i-n+1}^{n} k \right] \\
&= n! c_n + \sum_{i=1}^{\infty} \left[ c_{i+n} x^i \prod_{k=i}^{n+i} k \right],
\end{aligned}
\tag{24}
$$

where by the fact $c_i > 0 \forall i > 1$, we have that $\frac{\partial^n f(x)}{\partial x^n} \in \mathcal{F}$. Then by Proposition 1 we immediately proved the proposition. $\qquad\square$

### A.5 PROOF OF PROPOSITION 4

**Proposition 4.** *Let $f_1(x) = c_0^1 + \sum_{i=1}^{\infty} c_i^1 x^i \in \mathcal{F}$, and $f_2(x) = c_0^2 + \sum_{i=1}^{\infty} c_i^2 x^i \in \mathcal{F}$. Then for an adjancency matrix $\tilde{\mathbf{B}} \in \mathbb{R}_{\geqslant 0}^{d \times d}$ with spectral radius $\rho(\tilde{\mathbf{B}}) \leqslant \min(\lim_{i \to \infty} c_i^1/c_{i+1}^1, \lim_{i \to \infty} c_i^2/c_{i+1}^2)\}$, the following three statements are equivalent:*

1. *$\tilde{\mathbf{B}}$ forms a DAG;*

2. *$\operatorname{tr}[f_1(\tilde{\mathbf{B}}) + f_2(\tilde{\mathbf{B}})] = (c_0^1 + c_0^2)d$;*

3. *$\operatorname{tr}[f_1(\tilde{\mathbf{B}}) f_2(\tilde{\mathbf{B}})] = c_0^1 c_0^2 d$.*

*Proof.* By properties of analytic functions, we have

$$
f_1(x) + f_2(x) = c_0^1 + c_0^2 + \sum_{i=1}^{\infty} (c_i^1 + c_i^2) x^i
\tag{25}
$$

is an analytic function, and its convergence radius is given by

$$
\lim_{i \to \infty} (c_i^1 + c_i^2)/(c_{i+1}^1 + c_{i+1}^2) = \min(\lim_{i \to \infty} c_i^1/c_{i+1}^1, \lim_{i \to \infty} c_i^2/c_{i+1}^2),
\tag{26}
$$

and thus by Proposition 1 the statement 1 and 2 are equivalent. Similarly by properties of analytic functions statement 1 and 3 are equivalent. Thus the 3 statements are equivalent. $\qquad\square$

### A.6 PROOF OF PROPOSITION 5

**Proposition 5.** *Let $n$ be any positive integer, the adjacency matrix $\tilde{\mathbf{B}} \in \{\hat{\mathbf{B}} \in \mathbb{R}_{\geqslant 0}^{d \times d} | \rho(\hat{\mathbf{B}}) \leqslant s\}$ if and only if*

$$
\operatorname{tr}(\mathbf{I} - \tilde{\mathbf{B}})^{-n} = d,
$$

*and the gradients of the DAG constraints satisfies that $\forall \tilde{\mathbf{B}} \in \{\hat{\mathbf{B}} \in \mathbb{R}_{\geqslant 0}^{d \times d} | \rho(\hat{\mathbf{B}}) \leqslant s\}$*

$$
\|\nabla_{\tilde{\mathbf{B}}} \operatorname{tr}(\mathbf{I} - \tilde{\mathbf{B}})^{-n}\| \leqslant \|\nabla_{\tilde{\mathbf{B}}} \operatorname{tr}(\mathbf{I} - \tilde{\mathbf{B}})^{-n-k}\|,
$$

*where $k$ is an arbitrary positive integer, and $\|\cdot\|$ an arbitrary matrix norm.*

*Proof.* By Proposition 4 or Proposition 3, it would be straightforward that $\operatorname{tr}(\mathbf{I} - \tilde{\mathbf{B}})^{-n} = d$ is a necessary and sufficient condition for an adjacency matrix $\tilde{\mathbf{B}} \in \{\hat{\mathbf{B}} \in \mathbb{R}_{\geqslant 0}^{d \times d} | \rho(\hat{\mathbf{B}}) \leqslant s\}$ to form a DAG.

For the norm of gradients, it is straightforward that

$$\frac{\partial (1-x)^{-n}}{\partial x} = n(1-x)^{-n-1}. \tag{27}$$

For arbitrary $n$ we have

$$(1-x)^{-n} = 1 + \sum_{i=1}^{\infty} \left[ \prod_{j=n}^{n+i-1} j \right] x^i, \tag{28}$$

and obviously the coefficients is monotonic increasing w.r.t. $n$. Thus by the fact $\forall \tilde{\mathbf{B}} \in \{\hat{\mathbf{B}} \in \mathbb{R}_{\geqslant 0}^{d \times d} | \rho(\hat{\mathbf{B}}) \leqslant s\}$ we have for any $j > 0, k > 0$

$$\|(\mathbf{I} - \tilde{\mathbf{B}})^{-j}\| \leqslant \|(\mathbf{I} - \tilde{\mathbf{B}})^{-j-k}\|. \tag{29}$$

As a result, we have

$$\begin{aligned}
\|\nabla_{\tilde{\mathbf{B}}} \mathrm{tr}(\mathbf{I} - \tilde{\mathbf{B}})^{-n}\| =& n\|(\mathbf{I} - \tilde{\mathbf{B}})^{-n-1}\| \leqslant (n+k)\|(\mathbf{I} - \tilde{\mathbf{B}})^{-n-1} \\
\leqslant& (n+k)\|(\mathbf{I} - \tilde{\mathbf{B}})^{-n-k-1} = \|\nabla_{\tilde{\mathbf{B}}} \mathrm{tr}(\mathbf{I} - \tilde{\mathbf{B}})^{-n-k}\|.
\end{aligned} \tag{30}$$

$\square$

## B  ANALYZE NON-CONVEXITY OF DIFFERENT DAG CONSTRAINTS

In this section, we compare the non-convexity of different DAG constraints. Consider a DAG constraint (regularizer) derived from analytic function as follows:

$$\mathrm{tr} f(\tilde{\mathbf{B}}) = \mathrm{tr} \sum_i c_i \tilde{\mathbf{B}}^i, \tag{31}$$

we can analyze its convexity by analyzing its Hessian.

Firstly, the derivative of matrix power can be obtained using the following equation,

$$\nabla_{\tilde{\mathbf{B}}} \tilde{\mathbf{B}}^k = \sum_{j=0}^{k-1} \tilde{\mathbf{B}}^j \otimes \tilde{\mathbf{B}}^{k-1-j}, \tag{32}$$

where $\otimes$ denotes the Hardmard product. Thus the Hessian of (31) can be obtained as follows:

$$\nabla_{\tilde{\mathbf{B}}}^2 \mathrm{tr} f(\tilde{\mathbf{B}}) = \sum_{i=1}^{\infty} c_i \sum_{j=0}^{i-1} \left[\tilde{\mathbf{B}}^j\right]^\top \otimes \left[\tilde{\mathbf{B}}^{i-1-j}\right]^\top, \tag{33}$$

To exactly evaluate and analyze the Hessian exactly is highly non-trivial even for simplest function such as exponential if we do not have further assumption on matrix $\tilde{\mathbf{B}}$ (Magnus et al., 2021).

Obviously, the Hessian Equation (33) is symmetric and not positive semi-definite. One widely used way to convexify Hessian is to find a positive scalar $\eta$ such that

$$\Delta = \nabla_{\tilde{\mathbf{B}}}^2 \mathrm{tr} f(\tilde{\mathbf{B}}) + \eta \mathbb{I},$$

becomes positive semidefinite. Obviously, once $\eta$ is larger than the absolute value of largest Eigen value of $\nabla_{\tilde{\mathbf{B}}}^2$. Particularly for real symmetric matrices, the absolute value of largest Eigen value is the spectral norm, and it is up bounded by the Frobenius norm. Thus we can use the Frobenius norm of Hessian to roughly measure the non-convexity of a DAG constraints.

Besides the Frobenius norm, another way to obtain a lower bound of $\eta$ is to use the result that diagonal dominant matrix must be positive semi-definite. By the fact that all entries of $\hat{\mathbf{B}}$ is positive, obviously all entries of the Hessian are positive. In this case, a lower bound of $\eta$ convexify the problem is the maximum off-diagonal entries of the Hessian.

By Equation (33), by the fact all entries of $\tilde{\mathbf{B}}$ is positive, obviously at some point $\tilde{\mathbf{B}}$ the Frobenius norm, or the maximum off-diagonal entries of Hessian only depends on $c_i$. Thus it would be very straightforward that for the Order-{1,2,3,4} constraints used in our experiments, the higher order will results in stronger non-convexity. Furthermore, all these 4 constraints suffer from stronger non-convexity but gain less gradient vanishing than constraints with infinite convergence radius as they have large coefficients $c_i$.

## C    TIME COMPLEXITY OF DIFFERENT DAG CONSTRAINTS

In our methodology, we deliberately steer clear of employing general polynomials as the efficient evaluation of general polynomials of matrices is highly non-trivial. Instead, our focus centers on specific polynomials like the inverse function, which can be evaluated with commendable efficiency, demanding only $\mathcal{O}(d^3)$ computations. Remarkably, the efficiency extends to evaluating the gradient of the inverse function, a process also attainable within a $\mathcal{O}(d^3)$ framework. Meanwhile, for a general polynomial, it requires at least complexity $\mathcal{O}(d^4)$.

Besides the

## D    ADDITIONAL EXPERIMENTS AND HYPER PARAMETERS

In terms of hyper-parameters, our selection involves $\alpha = 0.1$, $\lambda_1 = 0.05$, and $T = 5$.

In this section, we provide additional experiments on linear SEM with equal variance and known ground truth scale. Additional experimental results on large scale graphs (including false positive rate and true positive rate) is provided in Figure 4 and Figure 5, which is consistent with results of SHD.

We also conducted small scale experiments on ER2, ER3, ER4, SF2, SF4 and SF4 graphs, and the results is provides in Figure 6 and 7, where the DAG constraints with finite convergence radius performs better than the infinite convergence radius one (*i.e.* the exponential based constraint), and all of them have similar performance.

In all experiments in this paper, for continuous based approaches we use exactly the same hyper parameter as Bello et al. (2022), for conditional independent test and score based approaches we use the default parameter in Causal Discovery Toolbox[2].

### D.1    ADDITIONAL EXPERIMENTS ON UNKNOWN SCALE DATA

To eliminate the influence of Signal-to-Noise ratio, we futher did an experiment, where the variance of noise on each node is randomly sampled from uniform distribution $U(1, 10)$ and the results is shown in the following table. The other protocal is the same as our original experiment.

Table 2: DAG learning performance of different algorithms on linear SEM with unknown ground truth scale (measuredby SHD of CPDAGs denoted by SHDC, the lower the better). The result is reported in the format of average$\pm$ standard derivation gathered from 10 different simulations.

|  | GES | PC | Exponential | Order-1 | Order-2 |
|---|---|---|---|---|---|
| 15-node ER2 (SHDC) | 27.1$\pm$11.0 | 26.0$\pm$8.1 | 25.6 $\pm$ 9.8 | 26.3 $\pm$ 10.1 | 26.9 $\pm$ 8.2 |
| 50-node ER1 (SHDC) | 23.9$\pm$7.34 | 14.1$\pm$7.8 | 25.5$\pm$8.0 | 29.3$\pm$10.7 | 37.0 $\pm$10.3 |

On 15-node ER2 graph, the exponential constraint attains better performance than PC and GES, while other two approaches performs worse. For 50-node ER1 graph, the exponential constraint attains comparable performance as GES, but worse than PC. These results are consistency with our intuition that for this kind of problem non-convexity is the main issue and constraints with finite spectral radius would suffer more from non-convexity.

---

[2]https://fentechsolutions.github.io/CausalDiscoveryToolbox/html/index.html

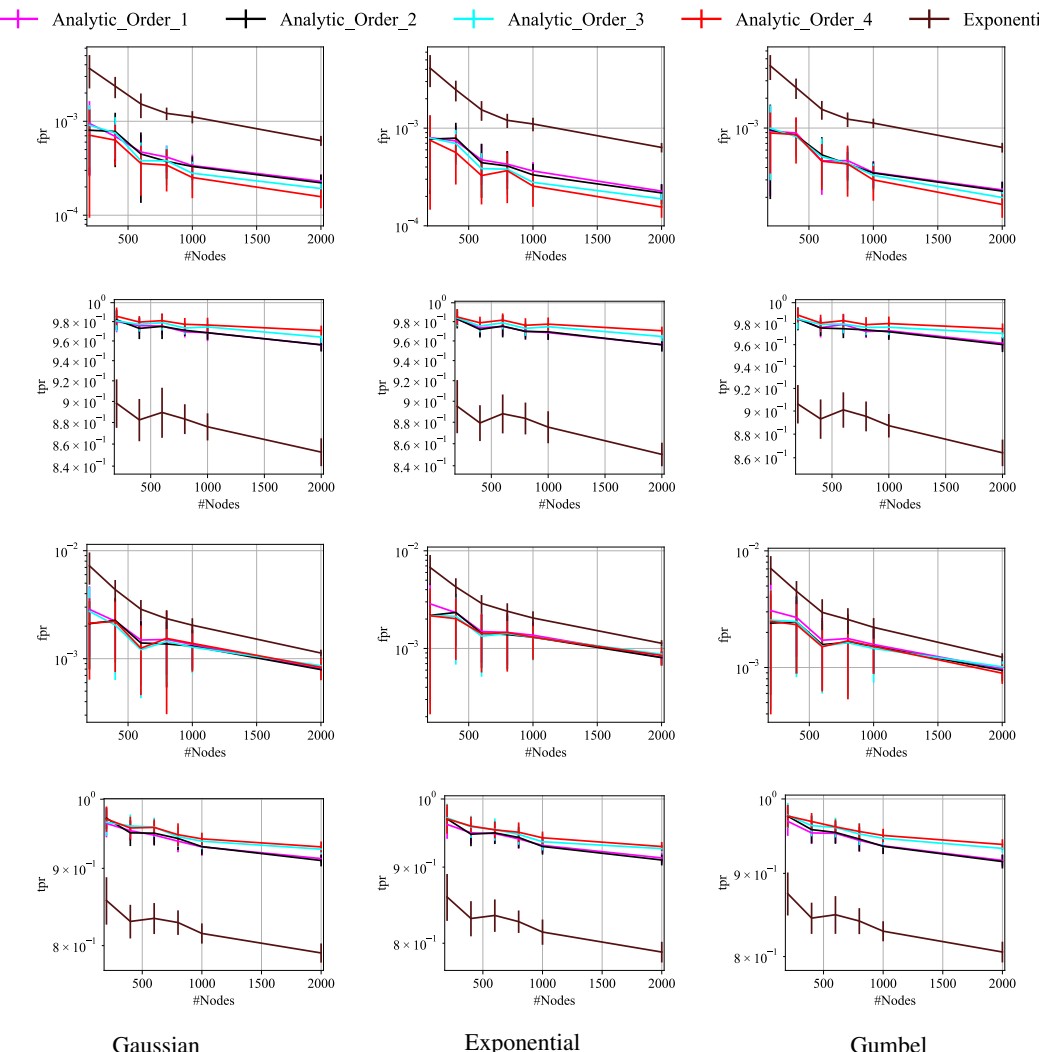

Figure 4: Additional experimental results on large scale ER3 (**Top 2 rows**) and ER4 (**Bottom 2 rows**) graphs. In terms of false positive rate (shown in the **First** and the **Third** row, the lower the better), all DAG constraints with finite convergence radius performs better than the Exponential based approach, and the Order-4 (*i.e.* the one with largest Gradient norm) usually performs better than others. In terms of true positive rate (shown in the Second and the **Forth** row, the higher the better), all DAG constraints with finite convergence radius performs better than the Exponential based approach, and the Order-4 (*i.e.* the one with largest Gradient norm) usually performs better than others.

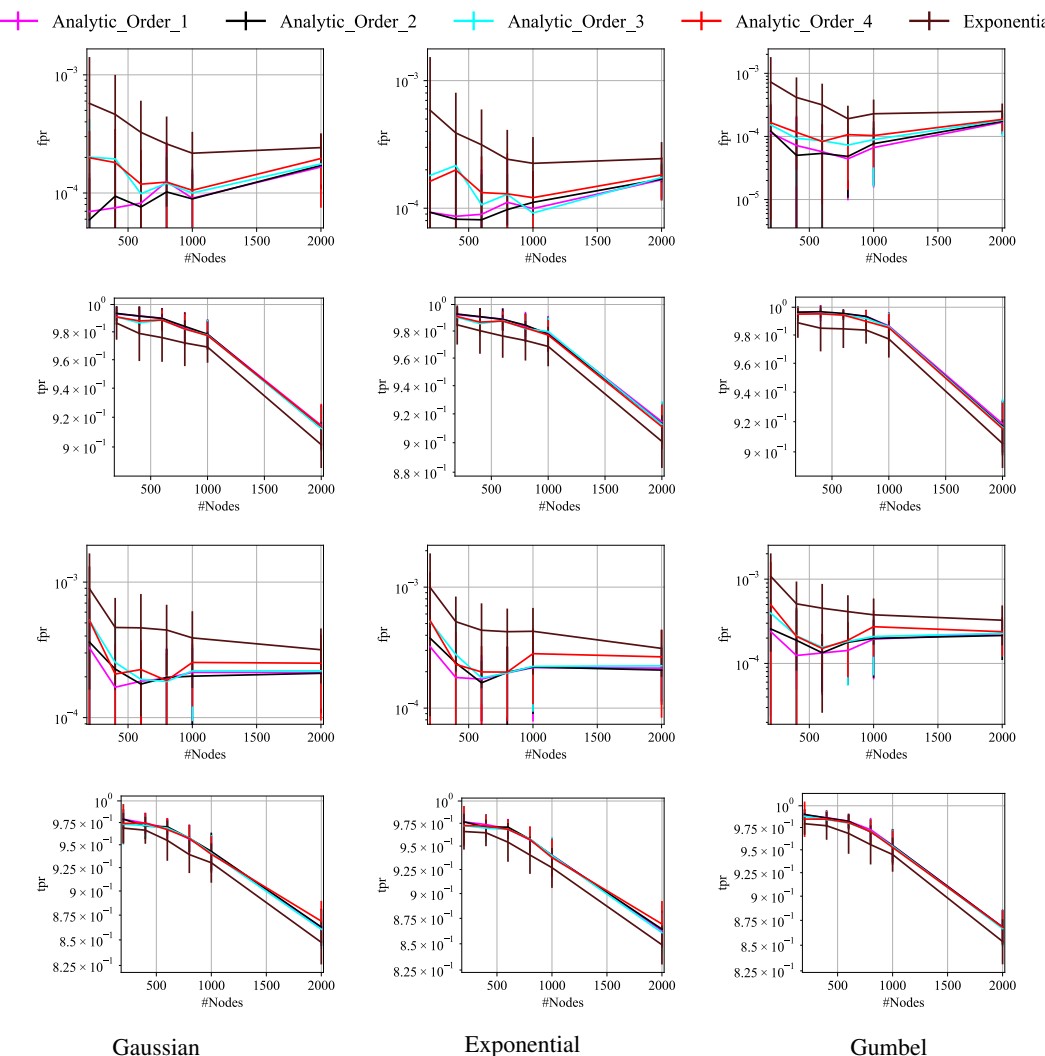

Figure 5: Additional experimental results on large scale SF3 (**Top 2 rows**) and SF4 (**Bottom 2 rows**) graphs. In terms of false positive rate (shown in the **First** and the **Third** row, the lower the better), all DAG constraints with finite convergence radius performs better than the Exponential based approach, and the Order-4 (*i.e.* the one with largest Gradient norm) usually performs a little bit worse others. In terms of true positive rate (shown in the **Second** and the **Forth** row, the higher the better), all DAG constraints with finite convergence radius performs better than the Exponential based approach, and the Order-4 (*i.e.* the one with largest Gradient norm) usually performs better than others.

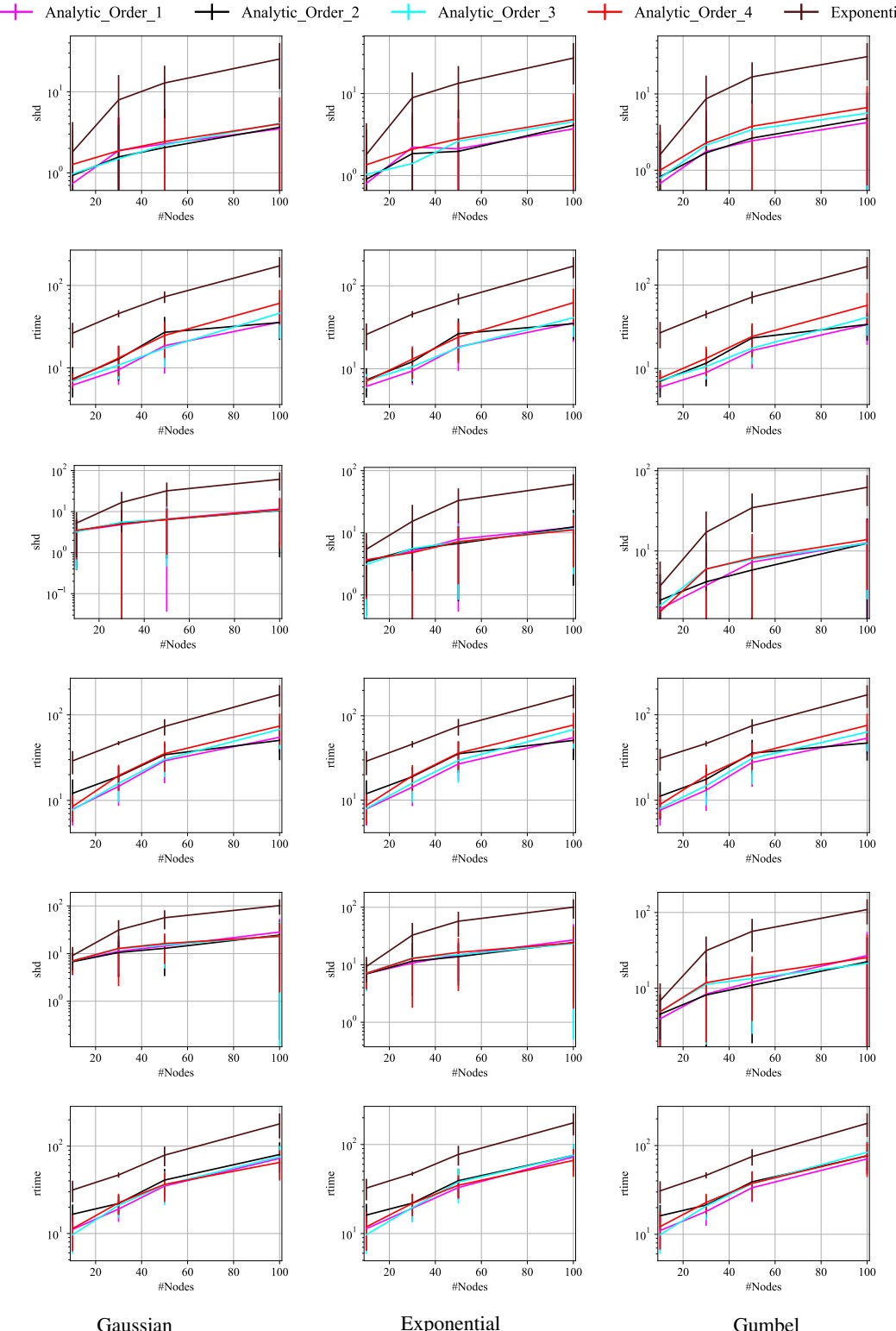

Figure 6: Additional experimental results on small scale ER2 (**Top 2 rows**), ER3 (**Top 2 rows**) and ER4 (**Bottom 2 rows**) graphs. In terms of SHD (shown in the **First**, the **Third** and the **Fifth** row, the lower the better), all DAG constraints with finite convergence radius performs better than the Exponential based approach, and they have similar performance. In terms of running time (shown in the Second, the **Forth** and the **Sixth** row, the lower the better), all DAG constraints with finite convergence radius performs better than the Exponential based approach, and they have similar performance.

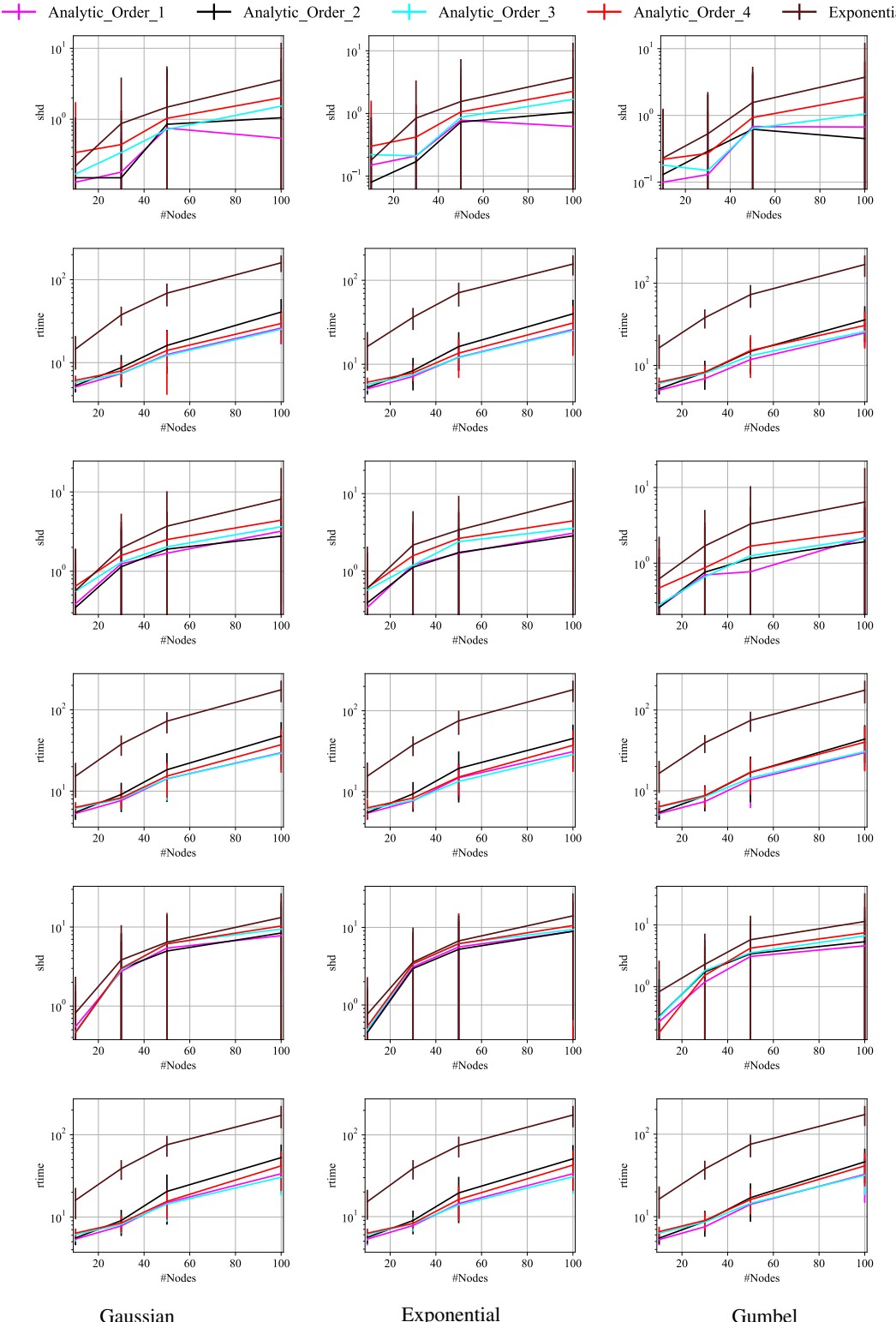

Figure 7: Additional experimental results on small scale SF2 (**Top 2 rows**), SF3 (**Top 2 rows**) and SF4 (**Bottom 2 rows**) graphs. In terms of SHD (shown in the **First**, the **Third** and the **Fifth** row, the lower the better), all DAG constraints with finite convergence radius performs better than the Exponential based approach, and they have similar performance. In terms of running time (shown in the Second, the **Forth** and the **Sixth** row, the lower the better), all DAG constraints with finite convergence radius performs better than the Exponential based approach, and they have similar performance.

