# OpenReview forum: "Analytic DAG Constraints for Differentiable DAG Learning"
_ICLR.cc/2024/Conference — Submitted to ICLR 2024_

### Official Review · Reviewer_JLDA · 2023-10-27

**Soundness:** 3 good
**Presentation:** 2 fair
**Contribution:** 3 good
**Rating:** 6
**Confidence:** 3

**Summary:**

The paper introduces a novel class of analytic functions, establishing that any element within this class can be utilized to construct a Directed Acyclic Graph (DAG) constraint. This family of functions demonstrates closure properties with respect to addition, multiplication, and differentiation. The author employs the properties of analytic function to explore the relationship between the phenomena of vanishing gradients and the convergence radius. On the empirical front, the author conducted two synthetic experiments to evaluate the performance of various DAG constraints. The findings highlight that the DAG constraint depends on the prior knowledge of data scale.

**Strengths:**

The paper presents a framework that integrates existing DAG constraints within the realm of analytic functions, employing the convergence radius as a analytical tool to investigate its relationship with the phenomenon of vanishing gradients. Upon a preliminary examination of the derivations, they appear to be sound and well-founded. While previous works have engaged in similar formulations to the proposed framework, the paper distinguishes itself through the analytic functions. The logical flow of the paper is commendable, facilitating ease of comprehension, although it is noted that there are certain aspects that remain ambiguous and warrant further clarification. Given the pivotal role of DAG constraints in the domain of structure learning, this work is poised to make a meaningful contribution.

**Weaknesses:**

There are two concerns about this paper. Firstly, the author mainly studied the effect of different DAG constraints with linear functional relationships. However, I wonder if those observations can be transferred to non-linear settings as well, where instead of using a weighted matrix $B$, one can directly apply the constraint on binary adjacency matrix, see [1]. Since non-linear behavior is ubiquitous in real world, the analysis on non-linear setting can further improve the contribution of this paper. Secondly, there are some ambiguities that requires further clarifications.


[1] Geffner, Tomas, et al. "Deep end-to-end causal inference." arXiv preprint arXiv:2202.02195 (2022).

**Questions:**

1. In the paper, "dataset scale" is an important concept but this has not bee properly introduced, what is the dataset scale and what do you mean be "dataset scale is known"? Do you provide extra information during model training?

2. For the DAG constraints, it seems that we only need summation order to be $d$ to specify a DAG. What are the advantages of going to $\infty$? Is it because $\infty$ order allows the series to converge to a particular function so that it is easy to compute the gradient?

3. In proposition, the $-n$ is for $(I-\tilde{B})$ or $tr(I-\tilde{B}))$? This can be quite misleading. For the previous discussion, I assume it is for $(I-\tilde{B})$?

4. Figure 1 is very helpful for the reader to understand the property of different DAG constraints. But the description of how figure 1 is generated is too vague, I think it would be helpful if the author can provide more details.

5. For "Choosing DAG constraints for different scenarios", I did not follow the arguments made in that section. For example, why with known data scale, the objective can provide a larger gradient? Also, if we have a large constraints, it will still create many local optima even with a informative objective function, right? So the correct argument is to achieve an appropriate balance between objective and constraints?

6. In experiment section, what is $\otimes$? Is it Kronecker product? Why do you use this instead of $\odot$?

7. For the experiment 4.1 and 4.2, why the dimensionality differs by a lot? For 4.1, the dimensionality starts at 200 but in 4.2, the highest is 50. I also want to see the performance of PC with known true data scale.

---

> ### Author Response · Authors · 2023-11-19
> **Rebuttal**
>
> ### In the paper, "dataset scale" is an important concept but this has not been properly introduced, what is the dataset scale and what do you mean be "dataset scale is known"? Do you provide extra information during model training?
>
> The data scale means the variance of the noise variable. For known scale problem, unnormalized raw data is provided and for unknown scale problem, normalized data is provided. During model estimation no other information is provided except for the unnormalized/normalized data. From the unnormalized data, we can get further information from the scale of variables while from normalized data we cannot get such information.
>
> ### For the DAG constraints, it seems that we only need summation order to be to specify a DAG. What are the advantages of going to $\infty$ ? Is it because order allows the series to converge to a particular function so that it is easy to compute the gradient?
>
> It would be easier to analyze the properties of DAG constraints when it goes to  $\infty$. For real complications, we may not need the power series. This is because the power series will converge to several specific functions which is easy to compute.
>
> ### Proposition 5 is not clear
>
> We have updated the equation to make it clear.
>
> ### Figure 1 and "Choosing DAG constraints for different scenarios"
> We have added a Section to discuss the non-convexity of the DAG constraints by using the Hessian of the DAG constraints. We roughly show that constraints with a finite spectral radius suffer more from non-convexity as they have larger $c_i$. However, they also have the benefits of less gradient vanishing as larger $c_i$ cause a larger norm of gradients. Thus for problems such as a problem with known scale, as the MSE loss provides enough information, the non-convexity may not be a serious issue and we can choose constraints with finite convergence radius regardless of its stronger non-convexity. Meanwhile, for a problem with an unknown scale, the MSE loss cannot provide enough guidance we will have to choose constraints with less non-convexity, i.e. constraints with infinite convergence radius.
>
>
> ### Typos $\otimes$ in experimental section
> We have fixed the typos.
>
>
> ### For the experiment 4.1 and 4.2, why does the dimensionality differ by a lot? For 4.1, the dimensionality starts at 200 but in 4.2, the highest is 50. I also want to see the performance of PC with known true data scale.
>
> Algorithms such as PC and GES are not scalable, to compare with them we have to use small graphs. The performance of PC is almost the same as the performance on unknown data scale, and it is worse than differentiable DAG learning used in our paper. This is widely reported in previous work. We have applied PC and GES on small graphs (up to 100 nodes) with known scale, and the result is consistent with previously reported ones, where PC and GES perform worse than continuous structure learning approaches. We will add these results in the future version.

---

> > ### Comment · Reviewer_JLDA · 2023-11-22
> >
> > Thanks for the authors' response regarding my questions.
> >
> > It address most of my concerns, I will keep my current rating.

---

### Official Review · Reviewer_EZWF · 2023-11-01

**Soundness:** 3 good
**Presentation:** 3 good
**Contribution:** 3 good
**Rating:** 6
**Confidence:** 3

**Summary:**

The authors propose an interesting framework that unifies previously proposed DAG constraints and proposes new ones. They also study constraints that differently from the popular one from NOTEARS mitigate the vanishing gradient (VG) problem. The main story is around the convergence radius of the analytic functions defining the constraints: a finite one mitigates VG but exacerbates nonconvexity, and with infinite radius it becomes viceversa.

**Strengths:**

- Interesting unifying framework for existing constraints
- Some guidelines provided for how to choose constraints in practice, for the case of linear SEMs with additive noise, equal variances

**Weaknesses:**

- First, the paper is overflowing the 9 pages limit ?
- What is the effect of the multiplication introduced in Eq 14 to get positivity ? This looks a bit hacky and wasn't done in previous related works ?
- The part "Choosing DAG constraints for different scenarios" is a bit too informal. Can you expand  the arguments more formally (if no space, at least in the appendix) ?

**Questions:**

No further questions beyond those in "weaknesses"

---

> ### Author Response · Authors · 2023-11-19
> **Rebuttal**
>
> We would like to thank Reviewer EZWF for insightful comments and detailed comments.
>
> ## Extension of non-linear cases
> The approach can be directly extended to a nonlinear case by replacing the DAG constraints in DAG-GNN, or nonlinear version of NOTEARS with ours. However, in the case of nonlinear problems, except for DAG constraints, there are many other factors that will affect the final result. We will discuss these problems in the future version.
>
> ## Paper exceeds 9 pages
> We are sorry for the problem. Due to limited time there was a technique problem, which results in a very large black area around Figure 2. This issue is now fixed.
>
> ## What is the effect of the multiplication introduced in Eq 14 to get positivity
> This is the standard approach in continuous DAG learning. This trick may have pros and cons, for more details we refer to Dennis Wei, Tian Gao, and Yue Yu. Dags with No Fears: A closer look at continuous optimization for learning
> bayesian networks. Advances in Neural Information Processing Systems, 33:3895–3906, 2020, and Ignavier Ng, Sébastien Lachapelle, Nan Rosemary Ke, Simon Lacoste-Julien, and Kun Zhang. On the convergence of continuous constrained optimization for structure learning. In International Conference on Artificial
> Intelligence and Statistics, pp. 8176–8198. PMLR, 2022.
>
> ## The part "Choosing DAG constraints for different scenarios" is a bit too informal. Can you expand the arguments more formally (if no space, at least in the appendix) ?
>
> We have now added a section, discussing the relationship between Hessian of DAG constraints and the convergence radius. The Hessian of the DAG constraints is symmetric but not positive semi-definite. One common way to convexify the problem is to add a term $\eta \mathbb{I}$ to the Hessian (equivalently, add $\eta ||\tilde{\mathbf{B}}||_F^2$) to the objective. If $\eta$ is larger than the Frobenius norm of the Hessian, the problem is convexified. Thus we can use the Frobenius norm of the Hessian to roughly measure the non-convexity of the DAG constraints.
>
>
>  The Frobenius norm of the Hessian, at some particular point $\tilde{\mathbf{B}}$, only depends on coefficients $c_i$ of the analytic function. From Eq (33) in the updated paper, it clearly shows that larger $c_i$ causes a large Frobenius norm of the Hessian, which roughly shows that constraints with a finite spectral radius suffer more from non-convexity as they have larger $c_i$. However, they also have the benefits of less gradient vanishing as larger $c_i$ cause a larger norm of gradients.

---

### Official Review · Reviewer_mZ7h · 2023-11-02

**Soundness:** 4 excellent
**Presentation:** 4 excellent
**Contribution:** 2 fair
**Rating:** 5
**Confidence:** 4

**Summary:**

The authors study a class of analytic functions that can be used as differentiable DAG constraints. They characterize the properties of the function class and show that it remains closed under various operations. They compare and contrast many existing DAG constraints with the analytic functions under the proposed class. They also study the tradeoff between the gradient vanishing and nonconvexity of the proposed constraints.

**Strengths:**

The paper presents a sound theoretical analysis of an analytic function class that can be used as differentiable DAG constraints. They also shed light on the gradient-vanishing issue encountered by the existing constraint-based methods such as [Bello et al., 2022] and [Zhang et al., 2022]. They propose a workaround albeit with a possibility of making the problem more nonconvex. The paper is well written and presented and the authors convey their ideas well.

**Weaknesses:**

The authors do well in comparing and contrasting their ideas with [Bello et al., 2022] and [Zhang et al., 2022]. Their observation is novel and provides an insight into the existing results, however, it seems like a natural extension. The numerical experiments with constraint-based methods do not suggest any major performance improvement over the existing methods (considering both shd and rtime). Furthermore, comparison with score-based methods also fails to show any major performance improvement.

**Questions:**

While I understand the intuitive tradeoff between the nonconvexity of the problem and the gradient-vanishing phenomena, is it possible to quantify such intuition? In my opinion, such a quantification would certainly improve the quality of the contributions. What is the mathematical meaning of more or less nonconvex and how does it relate to the DAG recovery in a formal way?

---

> ### Author Response · Authors · 2023-11-19
> **Rebuttal**
>
> We would like to thank Reviewer mZ7h for insightful comments and detailed comments. We have added a Section in the Appendix discussing the non-convexity using the Hessian of the DAG constraints. We established the quantified relation between the non-convexity of the problem and the Frobenius norm of the Hessian. Based on this, we can analytically conclude that DAG constraints with a finite convergence radius will suffer more from gradient vanishing but less from non-convexity. This is because large $c_i$ in the constraints will cause larger gradients but also results in a larger Hessian which causes stronger non-convexity.

---

> > ### Author Response · Authors · 2023-11-22
> > **Rebuttal**
> >
> > Dear Reviewer mZ7h,
> >
> > We deeply appreciate the attention and consideration you have given to our work. We genuinely hope that the provided clarification serves as valuable information, effectively addressing any concerns you may have had and contributing to an enhanced understanding of the merits and contributions of our work. Should you have additional questions or concerns, we welcome further discussion. We sincerely appreciate the time and effort you dedicated to reviewing our work.

---

> > > ### Comment · Reviewer_mZ7h · 2023-11-23
> > > **Thank you for your response**
> > >
> > > I have gone through the response and I have decided to keep my score unchanged.
> > >
> > > You should be careful with comparing nonconvexity using the Frobenious norm of the Hessian and would recommend strongly against adding an extra section about that. A simple example would be taking Hessian as $\begin{bmatrix}1 & 2 \\\\ 2 & 1 \end{bmatrix}$ and $\begin{bmatrix}2 & 1 \\\\ 1 & 2 \end{bmatrix}$. Clearly, the Frobenious norm would give you no information regarding the non-convexity (or convexity) of the problem here. If there are some other technical conditions that make this comparison possible in your case, then you should list those in the newly added section. Otherwise, it just confuses the readers more.

---

> ### Author Response · Authors · 2023-11-23
> **Rebuttal**
>
> We would like to clarify that for general matrix. It would not be a good idea to use Frobenious norm as a measurement of non-convexity. However, in our case the Hessian is known to be indefinite. In this case, we can add $\eta\mathbb{I}$ to the matrix to make it positive semi-definite. Here the Frobenious norm of the matrix can serve as a lower bound of $\eta$ (through may not be tight), that is if $\eta$ is larger than the Frobenius norm of the Hessian, then the sum of Hessian and $\eta\mathbb{I}$ must be  positive semidefinite. Considering this, in our case the Frobenious norm can be used as a rough measurement of non-convexity.
>
> Another way to make the Hessian positive semi-definite is to make it diagonally dominant. In that case, a sufficient condition for the sum of $\eta\mathbb{I}$ and the Hessian to be positive is $\eta$ is larger than the maximum off-diagonal entries of the Hessian. Again with similar analysis as Frobenious norm, we can show that the maximum entry of the Hessian will increase as the $c_i$ increases. Which suggests that constraints with finite convergence radius may suffer stronger non-convexity.
>
> Through both Frobenious norm and maximum off-diagonal  value of Hessian are lower bound of $\eta$ and they may not be tight, they can still serve as a rough measurement of the non-convexity of Hessian as the Hessian is known to be indefinite.

---

> > ### Author Response · Authors · 2023-11-23
> > **Rebuttal**
> >
> > In the given example, the second matrix is positive semi-definite and thus we can not use Frobenious norm to measure its non-convexity.

---

> > ### Author Response · Authors · 2023-11-23
> > **Rebuttal**
> >
> > In our new section, we have clearly stated that due to the indefinite property of the Hessian, we can then use Frobenious norm as a rough measure of the non-convexity.

---

> > > ### Comment · Reviewer_mZ7h · 2023-11-23
> > >
> > > Let $\lambda_1 > \lambda_2 > \lambda_3 > 0$. Consider a symmetric square matrix (Hessian) $A$ with eigenvalues $\lambda_1, \lambda_2, -\lambda_3$. It is easy to see that $A + \eta I$ has eigenvalues $\lambda_1 + \eta, \lambda_2 + \eta, -\lambda_3 + \eta$.  Furthermore, $\\| A \\|_F = \sqrt{\lambda_1^2 + \lambda_2^2 + \lambda_3^2}$.
> > >
> > > Observe:
> > > 1. $A$ is not positive semidefinite.
> > > 2. $\eta \geq \lambda_3$ makes it positive semidefinite.
> > > 3. $\\| A \\|_F$ can be increased by simply increasing $\lambda_1$ and this does not make $A$ more or less positive semidefinite in the sense that we still need  $\eta \geq \lambda_3$ to make it positive semidefinite.
> > >
> > > The larger point is that it might be misleading to use the Frobenious norm as a measure of nonconvexity irrespective of $A$ being positive semidefinite or not.

---

> > > > ### Author Response · Authors · 2023-11-23
> > > > **Rebuttal**
> > > >
> > > > Thanks for your comments.
> > > >
> > > > In the new added section, we have clearly stated that we are using Frobenious norm as an upper bound of the absolute value of the largest eigen value. Thus, it is a lower bound of $eta$ that can be used to form $\eta\mathbb{I}$ to convexify the problem. The lower bound, may not be a precise measure of non-convexity, but still can rough measure the non-convexity of the problem.
> > > >
> > > > Furthermore, in our case, the Hessian is a symmetric matrix, with all positive entries, and it is indefinite. In this case, the indefinity is mainly caused by the off-diagonal entries of the Hessian. Intuitively, the larger the off-diagonal entries will cause stronger non-convexity, and also the larger Frobenious norm.
> > > >
> > > > Besides Frobenious norm, the largest off-diagonal entry of the Hessian is also a lower bound of $\eta\mathbb{I}$, and it also increases as the increase of $c_i$ (this is provided in the updated revision). Though these measurements of non-convexity may not be precise as both bound may not be tight, they can still help to guide the choice of DAG constraints.

---

> > > > ### Author Response · Authors · 2023-11-23
> > > > **Rebuttal**
> > > >
> > > > The provided example would not occur in our case, as all entries of our Hessian are positive. We have stated in the new section that we are using Hessian as an upper bound of the largest absolute value of eigen values. This bound, may not be precise as it is not tight, but may still provide a good guidance for the choice of DAG constraints.

---

### Official Review · Reviewer_Y15A · 2023-11-02

**Soundness:** 3 good
**Presentation:** 2 fair
**Contribution:** 3 good
**Rating:** 6
**Confidence:** 4

**Summary:**

The paper provides a generalized framework for differentiable DAG learning (with observational data) with "DAGness" constraints, introducing a class of analytic functions that may act as DAG constraints and proving various properties and of such functions. In particular, the paper generalises a stream of recent works that includes NOTEARS, DAGMA and  TMPI.
Specific DAG constraints can be then derived picking functions in the identified class. The author suggest that the main factor of variation in terms of behavior of the resulting algorithm is determined by the radius of convergence $r$ of the analytic function: the two macro-classes being functions with finite and infinite $r$. The authors then suggest that there is a tradeoff between non-convexity (and potentially many local minimia) and gradient vanishing problems. The paper focuses on structural linear equation models.

**Important note:** the paper exceeds the 9 page limit and the final sections seem quite rushed to me. I reviewed the paper regardless of this, but must flag this issue as it may be unfair for other authors.

**Strengths:**

- I believe this is a solid contribution, in terms of results and clarity of exposition to the sub-class of methods for constraint-based DAG learning and linear SEMs: the suggested framework and results clearly subsume recent work in the area and can potentially constitute a solid ground for follow up research.
- The quality of the theoretical part of the paper is high: propositions and theorems are clearly stated and proofs are clear and easy to follow
- Notation is mostly well designed and background is sufficiently broad to make the paper self-contained for readers that have some knowledge in the field of DAG learning

**Weaknesses:**

- The clarity of the paper degrades after page 6. The final paragraph of the experimental section and the conclusions are rushed and need revision. Given that the rest of the paper is mostly clear and well written, this alone wouldn't be too much of a problem for me. However,  the paper also exceeds the 9 pages, and I think there is some potential for "unfair" comparison with other authors who spent time making sure to respect the 9 page limit (and polishing the entire submission).
- The weakest part of the work is in my opinion the connection between the convergence radius of the series and the trade-off vanishing gradient / non-convexity. I think the authors should elaborate more on this, especially for the part regarding the gradient vanishing.
-  I also had some difficulties following the discussion of the known-vs-unknown scale and how this relate to the convergence radius. The authors could elaborate more on it and also provide some (analytical) justifications that goes beyond intuitive arguments.
- The experiments only pertain synthetic data and do no report any comparison with score-based non-relaxed/discrete methods, see e.g. [1, 2, 3]

Minor comments/typos [excluding last sections, which need to be thoroughly revised]:
- Please define SEM the first time you introduce the acronym
- pag 3. An interesting property [of] the ....
- Check first sentence of sec 3.2
- pag 5, (probably typo) what's $b$ in $||Bb||_1$?
- Eq 16 shouldn't it be $B\circ B$ in the Frobenius norm?
- Is the term "invex" appropriate for describing the analytical constraints? Or maybe Fig 1 is misleading, as invex funcitons have stationary point $\implies$ global minimum.
- Probably would be more useful to show normalized SHD to account for growing DAG size.
- What's $\otimes$ on pag. 7?
- Please formulate an objective, or at least write down the score function for the modified problem at pag 9.
- Missing tr[...] of second line of Eq (22)?



References
[1] Nir Friedman and Daphne Koller. Being bayesian about network structure. A bayesian approach to structure discovery in bayesian networks. Machine learning, 50, 2003
[2] Bertrand Charpentier, Simon Kibler, and Stephan Günnemann. Differentiable DAG sampling. In
International Conference on Learning Representations, 2022
[3] Zantedeschi, Valentina, Luca Franceschi, Jean Kaddour, Matt J. Kusner, and Vlad Niculae. "DAG Learning on the Permutahedron." International Conference on Learning Representations, 2023

**Questions:**

See above.

---

> ### Author Response · Authors · 2023-11-19
> **Rebuttal**
>
> We would like to thank Reviewer Y15A for insightful comments and detailed comments. We now address the problems as follows.
>
> ### Paper exceed 9 pages, clarity after page 6.
>
> We are sorry for the problem. Due to limited time there was a technique problem, which results in a very large black area around Figure 2. This issue is now fixed. We will also carefully revise the experiment section to make it more clear.
>
> ### Connection between convergence radius and the trade-off vanishing gradient / non-convexity. Discussion of the known-vs-unknown scale.
> We have now added a section, discussing the relationship between Hessian of DAG constraints and the convergence radius. The Hessian of the DAG constraints is symmetric but not positive semi-definite. One common way to convexify the problem is to add a term $\eta \mathbb{I}$ to the Hessian (equivalently, add $\eta ||\tilde{\mathbf{B}}||_F^2$) to the objective. If $\eta$ is larger than the Frobenius norm of the Hessian, the problem is convexified. This means that we can use the Frobenius of the Hessian to roughly measure the non-convexity of the DAG constraints.
>
> The Frobenius norm of the Hessian, at some particular point $\tilde{\mathbf{B}}$, only depends on coefficients $c_i$ of the analytic function. From Eq (33) in the updated paper, it clearly shows that larger $c_i$ causes a large Frobenius norm of the Hessian, which roughly shows that constraints with a finite spectral radius suffer more from non-convexity as they have larger $c_i$. However, they also have the benefits of less gradient vanishing as larger $c_i$ cause a larger norm of gradients.
>
> For a problem with a known scale, it is well-known that the MSE loss provides sufficient information for estimating the graph structure, and thus the non-convexity would not cause too much problem as the MSE loss can help to pull the problem out of local minimum. Meanwhile, for a problem with an unknown scale, due to the symmetric property of the Gaussian distribution, the MSE loss, along with the DAG constraints, may suffer a lot from non-convexity as the MSE loss.
>
> ### The experiments only pertain synthetic data and do not report any comparison with score-based non-relaxed/discrete methods, see e.g. [1, 2, 3]
>
> Our algorithm is compared with a discrete score-based discrete method GES on a dataset with an unknown scale. Currently, most score-based non-relaxed/discrete approaches are not so scalable, and they only work on graphs with at most 100-200 nodes, meanwhile in most of our experiments the graph size can go up to 2000. In our revised version, we will add an experiment comparing with those methods on small graphs.
>
> ### Invex DAG constraints
> If we only consider the DAG constraints, it is invex. In the overall framework we further have constraints on the $\ell_1$ norm of the adjacency matrix. In this case, the curve of the DAG constraints becomes similar to Figure 1 due to the norm constraints. We will clarify this in the revised version.
>
> ### Eq (16)
> In Eq (16), $\mathbf{B}$ is the adjacency matrix so that it appears in the Frobenius norm. The DAG constraints only work for a positive weighted adjacency matrix, and the weight in $\mathbf{B}$ can be either positive or negative. Meanwhile, $\mathbf{B}$ is a DAG if and only if $\mathbf{B}\odot \mathbf{B}$ is a DAG. Based on this we apply DAG constraints on $\mathbf{B}\odot \mathbf{B}$.
>
> ### Other minor comments:
> We will carefully do a proofreading and fix all typos.

---

> > ### Comment · Reviewer_Y15A · 2023-11-20
> > **Thanks for your reply**
> >
> > I have read the rebuttal and I thank the authors for their reply and clarifications. I am satisfied with the rebuttal and will keep my original score. Please note that methods mentioned in the review are substantially dissimilar to GES and could possibly scale to higher dimensions; I appreciate the authors will carry out a comparisons with this class of methods as well.

---

### Author Response · Authors · 2023-11-19
**Rebuttal**

We would like to thank all reviewers for insightful comments and constructive suggestions. We have updated the draft according to the comments and we have addressed the raised questions individually.

---

### Meta-Review · Area_Chair_6HVV · 2023-12-11

**Metareview:**

This paper discusses on constraints that can be used for improving the learning of linear DAGs such as SEMs. However, experiments are not fully convincing and thus the impact has been questioned. Comparisons with other approaches could be included, as suggested in the reviews. The discussion on (non-)convexity has been confusing, as acknowledged by the authors. While it might (or not) be all fine, the raised doubts have not been fully resolved yet.

**Justification For Why Not Higher Score:**

Some discussions have left doubts. Comparisons have been considered only partly good for highlighting the benefits of the new approach, so the amount of novelty/incremental work has been put in consideration too. In the end, all committee members are borderline about the submission.

**Justification For Why Not Lower Score:**

N/A

---

### Decision · Program_Chairs · 2024-01-16

Reject